# Adjuvant Oligonucleotide Vaccine Increases Survival and Improves Lung Tissue Condition of B6.Cg-Tg (K18-ACE2)2 Transgenic Mice

Volodymyr V. Oberemok [1,*] , Kateryna V. Laikova [1], Kseniya A. Yurchenko [1] , Ilya A. Novikov [1] ,
Tatyana P. Makalish [2], Anatolii V. Kubyshkin [2] , Oksana A. Andreeva [1] and Anastasiya I. Bilyk [1]

[1] Department of Molecular Genetics and Biotechnologies, Institute of Biochemical Technology, Ecology and Pharmacy, V.I. Vernadsky Crimean Federal University, Simferopol 295007, Crimea; botan_icus@mail.ru (K.V.L.); yurchenkokseniya28@gmail.com (K.A.Y.); i.nowikow2012@mail.ru (I.A.N.); andreeva-oksana.94.3@mail.ru (O.A.A.); bilyk.ai97@mail.ru (A.I.B.)

[2] Medical Academy Named after S.I. Georgievsky, V.I. Vernadsky Crimean Federal University, Simferopol 295007, Crimea; gemini_m@list.ru (T.P.M.); kubyshkin_av@mail.ru (A.V.K.)

* Correspondence: voloberemok@gmail.com; Tel.: +7-(978)-8146866

**Abstract:** The main problem in creating anti-coronavirus vaccines that target mainly proteins of the outer membrane of the virus is the rapid variability in the RNA genome of the pathogen that encodes these proteins. In addition, the introduction of technologies that can affordably and quickly produce flexible vaccine formulas that easily adapt to the emergence of new subtypes of SARS-CoV-2 is required. Universal adjuvant oligonucleotide vaccines based on conserved regions of the SARS-CoV-2 genome can take into account the dynamics of rapid changes in the virus genome, as well as be easily synthesized on automatic DNA synthesizers in large quantities in a short time. In this brief report, the effectiveness of four phosphorothioate constructs of the La-S-so-type adjuvant oligonucleotide vaccine is evaluated on B6.Cg-Tg (K18-ACE2)2 transgenic mice for the first time. In our primary trials, the oligonucleotide vaccine increased the survival rate of animals infected with SARS-CoV-2 and also reduced the destructive effects of the virus on the lung tissue of mice, activating both their innate and adaptive immunity. The obtained results show that the development of adjuvant oligonucleotide vaccine constructs of the La-S-so type is an affordable and efficient platform for the prevention of coronavirus infections, including those caused by SARS-CoV-2.

**Keywords:** adjuvant oligonucleotide vaccine; SARS-CoV-2; phosphorothioate oligonucleotides; innate immunity; adaptive immunity

## 1. Introduction

As of 10 March 2023, according to the Johns Hopkins University Coronavirus Resource Center, more than 676 million people had fallen ill from coronavirus, more than 6.8 million had died, and more than 13.3 billion vaccine doses had been administered worldwide [1]. The COVID-19 pandemic is now considered to be over [2]. However, outbreaks of the disease continue to be recorded throughout the world [3–5]. Obviously, the microevolution of SARS-CoV-2 will continue, and new subtypes will arise that can eventually lead to the next pandemic [6]. Due to the lack of specific drugs that can provide reliable protection, vaccines are an effective tool for preventing COVID-19 in this situation. To date, there are three main platforms for vaccines: inactivated [7], adenoviral vector [8], and mRNA vaccines [9]. However, all three platforms have side effects on human health. For example, the inactivated CoronaVac vaccine causes pain at the injection site, headache, fatigue, and muscle and joint pain [10]. Adenovirus vaccines, using VaxZevria as an example, cause frequent systemic reactions in the form of fatigue, myalgia, headache, fever, and atypical thrombosis [11,12]. Glomerulonephritis is reported to be one of the most serious side effects

following Moderna injections [13]. In addition, all of the listed vaccine platforms do not provide a long enough operational half-life for the rapidly changing SARS-CoV-2 RNA genome. A promising way to create anti-coronavirus vaccines is to include conservative regions of the coronavirus genome in universal vaccines [14]; however, this issue is still under study. Thus, from the point of view of improving the safety and duration of vaccines, the search for new platforms is relevant. Of interest is the post-genomic platform for creating vaccines [15], which is based on the use of nucleotide sequences of RNA viruses not just as adjuvants (based typically on phosphorothioate oligonucleotides containing CpG islands) but also as antigens [14–16]. The avalanche accumulation of SARS-CoV-2 genomic sequences contributes to this direction of the search.

For the first time, in this brief report, we will try to answer the question of the viability of nuclease-resistant adjuvant oligonucleotide vaccines with a phosphorothioate backbone using a practical example. For this study, we use B6.Cg-Tg (K18-ACE2)2 transgenic mice and four 'La-S-so' constructs containing an antigen-presenting 'head' with a specific sequence in order to activate adaptive immunity and a 'tail' with CpG islands to activate innate immunity. In addition, in the article, we consider two routes of administration: subcutaneous and intranasal. Our emphasis is on the survival of mice, weight gain, lung organometric parameters, and the state of the lung tissue. The fundamental possibility of activating innate and adaptive immunity by adjuvant oligonucleotide vaccines will be discussed.

## 2. Materials and Methods

### 2.1. Design, Synthesis, and Purification of a La-S-so-Type Adjuvant Oligonucleotide Vaccine

We used 4 phosphorothioate oligonucleotide constructs for the experiments:
La-S-so-1
5′-(CCCCCGGGGG)$_{\text{'neck'}}$(GCAGAGACAGAAGAAACAGCAAAC)$_{\text{'head'}}$ (CCCCCGGGGG)$_{\text{'neck'}}$(AACGCCAACGCC)$_{\text{'tail'}}$-3′;
La-S-so-2
5′-(CCCCCGGGGG)$_{\text{'neck'}}$(AGGCACAACAACAAGGCCAAAC)$_{\text{'head'}}$ (CCCCCGGGGG)$_{\text{'neck'}}$(AACGCCAACGCC)$_{\text{'tail'}}$-3′;
La-S-so-3
5′-(CCCCCGGGGG)$_{\text{'neck'}}$(AACAAGACAAAAACACCCAAGAAG)$_{\text{'head'}}$ (CCCCCGGGG)$_{\text{'neck'}}$(AACGCCAACGCC)$_{\text{'tail'}}$-3′;
La-S-so-4
5′-(CCCCCGGGGG)$_{\text{'neck'}}$(CACCGAGGCCACGCGGAG)$_{\text{'head'}}$ (CCCCCGGGGG)$_{\text{'neck'}}$(AACGCCAACGCC)$_{\text{'tail'}}$-3′.

The vaccine was designed using SARS-CoV-2 genomes, the GenBank database was used to search conserved sequences for 'head' regions inside La-S-so constructs, and the ClustalW 2.0.3 programs were used to align oligonucleotide sequences [17]. In our 'La-S-so' construct, we placed the CpG motifs in the 'tail' region in the 5′-purine-purine-unmethylated deoxycytosine-deoxyguanosine-pyrimidine-pyrimi- dine-3′ position [18]. The 'neck' region of the construct features a double-stranded DNA region stabilized by 30 hydrogen bonds between 10 cytosines and 10 guanines. The double-stranded DNA 'neck' makes it possible to form a loop in the region of the antigen-presenting 'head'. The general look of the La-S-so-type vaccine was represented in [14]. Only adenines, cytosines, and guanines are present in the construct, which makes it possible to activate pattern recognition receptors interacting with both 'non-self' DNA and 'non-self' RNA [19]. Since the PS backbone is the basis of the construct, the presence of deoxyribose is of little importance, and everything depends on the nucleotide sequence of the construct [20]. Moreover, CpG motifs in nuclease-resistant PS backbones have been found to dramatically enhance B cell stimulatory properties [21,22].

The developed vaccine was synthesized on the ASM-800 DNA synthesizer (BIOSSET, Novosibirsk, Russia) using standard phosphoramidite synthesis procedure. The synthesis was carried out in the direction from the 3′ to the 5′ end. After completion of all cycles of

synthesis, the target oligonucleotide construct was removed from the solid-phase support; the removal of the protective groups was carried out overnight at 55 °C in a concentrated ammonia solution (analytical grade, "Vekton", Saint Petersburg, Russia) [23]. Purification of the synthesized vaccine was performed by reverse-phase high-performance liquid chromatography (RP-HPLC) on a Jupiter 5 μm C18 300 Å column (4.6 mm × 250 mm) using a preparative HPLC system (Azura P6.1 L, UVD 2.1S detector) with DMT-on mode. Buffers used for purification were as follows: (A) 0.1 M triethylammonium acetate in water, and (B) 50% MeCN/buffer A with a gradient of 30%. After HPLC purification, DMT-on oligonucleotide peak was collected, and DMT groups were eventually eliminated using 80% aq. Acetic acid (analytical grade, "Vekton", Russia) for 20 min at 20 °C, followed by the evaporation of the acid. Oligonucleotide constructs were dissolved In Milli-Q water (Merck Millipore, Burlington, NJ, USA), and precipitation of oligonucleotide constructs with isopropanol was carried out.

### 2.2. Animals and Experimental Groups

Transgenic mice (B6.Cg-Tg (K18-ACE2)2) (The Jackson Laboratory, Bar Harbor, ME, USA) at the age of 7 weeks were randomly divided into 4 groups as follows: (1) intact; (2) with SARS modeling without treatment; (3) two intranasal injections of the vaccine with an interval of a week, one and two weeks before the SARS modeling; and (4) subcutaneous administration of the vaccine with an interval of a week, one and two weeks before the SARS modeling. There were six mice in each group. Experiment was performed in duplicate. The animals were kept in specific pathogen-free cages inside laminar flow hoods of the 2nd class of protection with free access to water and food.

### 2.3. Vaccination

The mixes of 4 adjuvant oligonucleotide vaccine constructs dissolved in Milli-Q water (Merck Millipore, Burlington, MA, USA) were administered intranasally and subcutaneously twice, at weekly intervals, with an amount of 30 μL at a concentration of 1000 ng/μL (250 ng of each La-S-so construct in microliter). Injections were carried out using an insulin syringe, and in the case of intranasal administration, the vaccine was administered intranasally with a pipette dispenser. During intranasal administration, mice were immersed in anesthesia, the vaccine was added to their nasal passages for further reflex inhalation, and the vaccine entered their respiratory tracts. All mice weighed 15.3 ± 0.2 at the beginning of the experiment (administration of the 1st dose of the adjuvant oligonucleotide vaccine).

### 2.4. Weighing

The weight of mice was measured every day on an Ad100 scale (Axis, Warsaw, Poland) with a precision of 0.001 g.

### 2.5. Mice Infestation

SARS was modeled by intranasal administration of a suspension of virus particles. For RNA isolation, 100 μL of a swab in 0.9% NaCl solution was used, and then 3.5% of the isolated RNA was used for RT-PCR. Biomaterial from 15 patients (2 μL each) with PCR-confirmed SARS-CoV-2 was mixed with a 0.9% NaCl solution, after which it was injected into mice in an amount of 30 μL of liquid, and thus all animals received relatively equal doses in terms of virulence. When using the Polivir SARS-CoV-2 Express kit (Litekh, Moscow, Russia), according to the instructions, Cq in these patients was always in the range from 17 to 18 cycles.

### 2.6. PCR Procedure

The infection of all mice was confirmed by PCR 2 days after SARS modeling. A swab for PCR analysis was taken from the oral cavity of mice. The resulting smear was placed in a test tube with a 0.9% NaCl solution. RNA was isolated using the "Polivir SARS-CoV-2

Express" kit (Litekh, Russia) according to the instructions. RT-PCR was also performed using "Polivir SARS-CoV-2 Express" kit (Litekh, Russia) according to the instructions. The quantitative PCR was carried out on CFX96 Touch (Bio-Rad, Hercules, CA, USA).

*2.7. Autopsy of Model Animals*

After SARS modeling, in every group, 2 mice were euthanized for organometric and immunohistochemical investigations on 5th, 10th, and 30th days. Animals were euthanized by intramuscular injection of chloral hydrate at a dose of 0.2 mg/g of live weight, after which they were decapitated.

The autopsy was carried out in several stages. First, the limbs were fixated with dissection needles on the dissecting tray with the animal's back down. Next, the skin was captured and lifted with tweezers; a longitudinal incision was made along the white line of the abdomen from the inguinal region to the region of the lower jaw. The abdominal wall remained intact. The skin was pushed back along the edges of the abdominal wall and fixed. An incision was made in the abdominal wall from the inguinal region to the midline of the abdomen, and an incision was made in the chest with scissors. After that, the lungs were carefully removed, and the target organs were placed in histological cassettes. Organs were stored in 10% buffered formalin for 12 h to prepare for immunohistochemical analysis.

*2.8. Immunohistochemical Assay*

Dehydration and paraffin impregnation were carried out in a Logos microwave histoprocessor (Milestone, Sorisole, Italy). Tissue blocks were sectioned into slices 4 μm thick and stained with hematoxylin and eosin. The following reagents for fixation, dehydration, wiring, and staining were manufactured by Biovitrum (Russia) and recommended for histological studies: formalin 10%, isopropyl alcohol, paraffin, o-xylene, and a set of dyes of hematoxylin and eosin. Stained sections were scanned on an Aperio CS2 scanner (Leica, Wetzlar, Germany) for morphometric analysis of changes in the lungs, 10 fields of view in each section were selected in the AperioScanScope program, and the relative areas occupied by the lumen of the alveoli, stroma, vessels, areas of edema, and fibrosis were calculated.

*2.9. Statistical Analysis*

Statistical calculations were carried out using the program STATISTICA 10. The normality of the distribution of the trait was determined by the Shapiro–Wilk method. The mean values of indicators, standard deviation, and error of the mean for the upper and lower quartiles were calculated. The Kruskal–Wallis method was used to compare scores between the groups. Differences were considered significant at $p < 0.05$.

The study was approved by the Ethics committee of the Federal State Autonomous Educational Institution of Higher Education V.I. Vernadsky Crimean Federal University, dated 1 October 2021, protocol No. 25/21. When conducting experimental studies, the principles and provisions of the Guide for the Care and Use of Laboratory Animals (USNIH, No. 85-23), international rules "Guide for the Care and Use of Laboratory Animals" (2009), and the Council of Europe Convention for the Protection of Vertebrate Animals used for Experimental or Other Scientific Purposes (Strasbourg, 1986) were taken into consideration.

**3. Results**

The adjuvant oligonucleotide vaccine increased the survival of transgenic mice (B6.Cg-Tg (K18-ACE2)2) infected with SARS-CoV-2 (Figure 1a). Between the 10th and 30th day, all the infected mice died in the group with the SARS modeling without treatment, while in the groups with the administered adjuvant oligonucleotide vaccine, the mice showed 100% viability.

In the course of the studies, a significant decrease in weight gain of the animals of the infected control group (the SARS modeling without treatment) was found during the 10 days of the experiment compared with that of the individuals from the groups in which

vaccination was carried out and the intact group (Figure 1b–d). Weight loss is a non-specific pattern in viral diseases, including COVID-19 [24–27]. Between the 2nd and 10th day of the experiment, a significant increase ($p < 0.05$) in the weight gain between the groups with the administered vaccine and the group with the SARS modeling without treatment (the infected control) was found. On 10th day, the average weight of the mice was $32.1 \pm 2.1$, $21.7 \pm 0.2$, $30.4 \pm 2.4$, and $29.9 \pm 1.9$ in the intact group, in the group with the SARS modeling without treatment, in the group with the SARS modeling and intranasal administration of vaccine, and the group with the SARS modeling and subcutaneous administration of vaccine, respectively. Of note, we also evaluated the influence of the intranasal and subcutaneous administration of the adjuvant oligonucleotide vaccine on weight gain in mice and found that the administration of the vaccine insignificantly increased the weight of the mice by 10–15% in comparison with the intact group after 14 days of the pre-SARS modeling period. The most pronounced increase in weight was found for the intranasal vaccine administration. It can be concluded that the applied mix of the four constructs of the La-S-so-type adjuvant oligonucleotide vaccine was able to save the animals from weight loss and death, which is consistent with the data of the organometric parameters and histological structure of the lungs of the mice (Table 1, Figure 2).

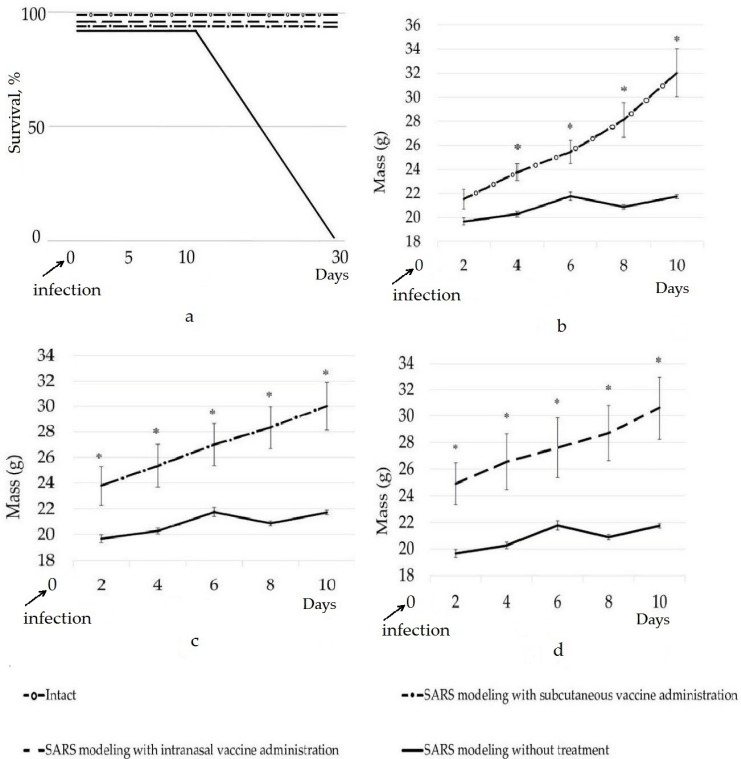

**Figure 1.** Dynamics of survival (**a**) and weight gain (**b–d**) of mice in the groups of the experiment; * significant difference compared to the control group (SARS modeling without treatment), ($p < 0.05$).

The analysis of the organometric parameters of the lungs showed significant changes in the area of various parts of the lungs (Table 1). The most pronounced dynamics of these changes were revealed in the study of three parameters of tissue sections: areas of edema, vessels, and the lumen of the alveoli. Interalveolar and perivascular edema in all groups became significant on the 5th day, but decreased on the 10th day and subsided by the 30th day; its area became significantly lower relative to that in the early stages of the disease. Congestion and the vascular area are the most pronounced in the group with the intranasal administration of the vaccine. At the beginning of the disease on the 5th day, in all groups, the greatest dilatation of capillaries was observed. On the 10th day in all groups (with and without treatment), alveolar-hemorrhagic syndrome was noted in about half of the animals. The area of the lumen of the alveoli in the animals was significantly

reduced due to distlectasis and atelectasis on the 10th day of illness in all groups, except for the group with the intranasal administration of the vaccine. It is this indicator that is the key, in our opinion, since it is the most pronounced in the group without treatment and is clinically accompanied by lethality, noted only in this group.

**Table 1.** Organometric parameters of the lungs of mice in the different groups of the experiment.

| Area % → Group | | Stroma | Edema | Vessels | Lumen of the Alveoli | Fibrosis |
|---|---|---|---|---|---|---|
| Intact (i) | | 40.0 [34.0;42.0] | 0 [0.0;0.0] | 12.50 [10.0;14.0] | 47.50 [43.0;50.0] | 0 [0.0;0.0] |
| Intranasal vaccine (n) | 5 day | 30.50 + [27.5;34.0] | 7.50 + [50;9.0] | 21.50 * [14.0;28.5] | 37.00 [31.5;44.5} | 1.50 [1.0;3.0] |
| | 10 day | 40.00 □ [37.0;45.0] | 5.00 [3.0;7.0] | 19.50 [13.0;31.0] | 30.50 * [25.0;41.0] | 1.00 [0.0;1.0] |
| | 30 day | 41.50 □ [39.0;45.5] | 2.00 □ [1.0;4.0] | 13.50 [9.5;17.0] | 42.00 [35.0;45.0] | 1.00 [0.0;1.0] |
| Subcutaneous vaccine (s) | 5 day | 35.00 [32.0;38.5] | 8.00 + [6.0;12.5] | 17.00 [12.0;21.5] | 39.50 [34.5;42.5] | 1.00 [1.0;2.0] |
| | 10 day | 45.50 [42.0;48.0] | 8.00 [8.30;13.0] | 17.50 [17.0;23.0] | 25.50 * [24.0;29.0] | 2.00 [2.0;3.0] |
| | 30 day | 39.50 [36.0;47.5] | 3.00 □ [1.5;4.0] | 16.50 [15.5;22.5] | 37.50 □ [32.5;41.0] | 1.00 [1.0;1.0] |
| SARS modeling without treatment (c) | 5 day | 40.50 [37.5;46.5] | 5.00 [4.0;8.5] | 15.00 [11.0;27.0] | 32.50 [22.0;41.5] | 1.00 [0.0;1.5] |
| | 10 day | 43.50 [38.0;50.0] | 8.00 [5.0;10.0] | 25.00 [21.0;28.0] | 20.00 * [19.0;25.0] | 1.50 [1.0;2.0] |
| | 30 day | x | x | x | x | x |

Note: differences between groups were determined by comparing independent samples using the Kruskal–Wallis method and were considered significant at $p < 0.05$; *—differences in comparison with the intact group; +—differences in compared to the group with SARS modeling without treatment at the corresponding time of the experiment; □—differences within the corresponding group at different periods of the experiment (compared to the 5th day); x—for this variant, the study was not carried out due to the complete death of mice.

For the group with the SARS modeling without treatment, the histochemical studies (Figure 2) showed significant areas of dyslectasis and atelectasis in the lung parenchyma, the thickening of the interalveolar septa, the edema and lymphocyte infiltration of the alveoli and bronchi, and the desquamation of the epithelium of the bronchioles on the 5th day after infection. By the 10th day, the proliferation of type II alveolocytes developed, fibrin clots appeared in the lumen of the vessels, hemorrhages were recorded in the lumen of the alveoli, and single hemosiderophages occured. Between the 10th and 30th days of the disease, all animals from this group died at night, and this did not allow us to study how the changes in the structure of the lung are associated with the development of SARS, since the autolysis additionally destroyed the tissues.

In the mice of the group with the subcutaneous administration of the vaccine, on the 5th day of the development of the disease, the edema of the interalveolar septa, areas of emphysema, extensive areas of lymphoid infiltration, hemorrhages, and a plethora of large vessels and capillaries were found. On the 10th day, the area of the lumen of the alveoli decreased due to atelectasis and the proliferation of alveolocytes, the vessels remained full-blooded, and areas of fibrosis and fibrin fibers appeared in the lumen of the vessels. By the 30th day, the area of the lumen of the alveoli slightly increased due to compensatory emphysema, while the state of the parenchyma as a whole remained the same.

The administration of the intranasal vaccine showed the best effect in assessing the morphology of the lung tissue of the experimental animals. At the initial stage (up to 5 days), the course of the disease turned out to be more acute than in other groups and was accompanied by significant edema in the interalveolar septa and paravasal region, lymphoid infiltration, and areas of atelectasis. On the 10th day, the edema became less noticeable, but the complete or partial collapse of the alveoli was observed. The vessels still remained full-blooded, and single hemorrhages were revealed. By the 30th day, no edema was detected; however, the interalveolar septa remained thickened due to lymphoid infiltration and the proliferation of lining cells.

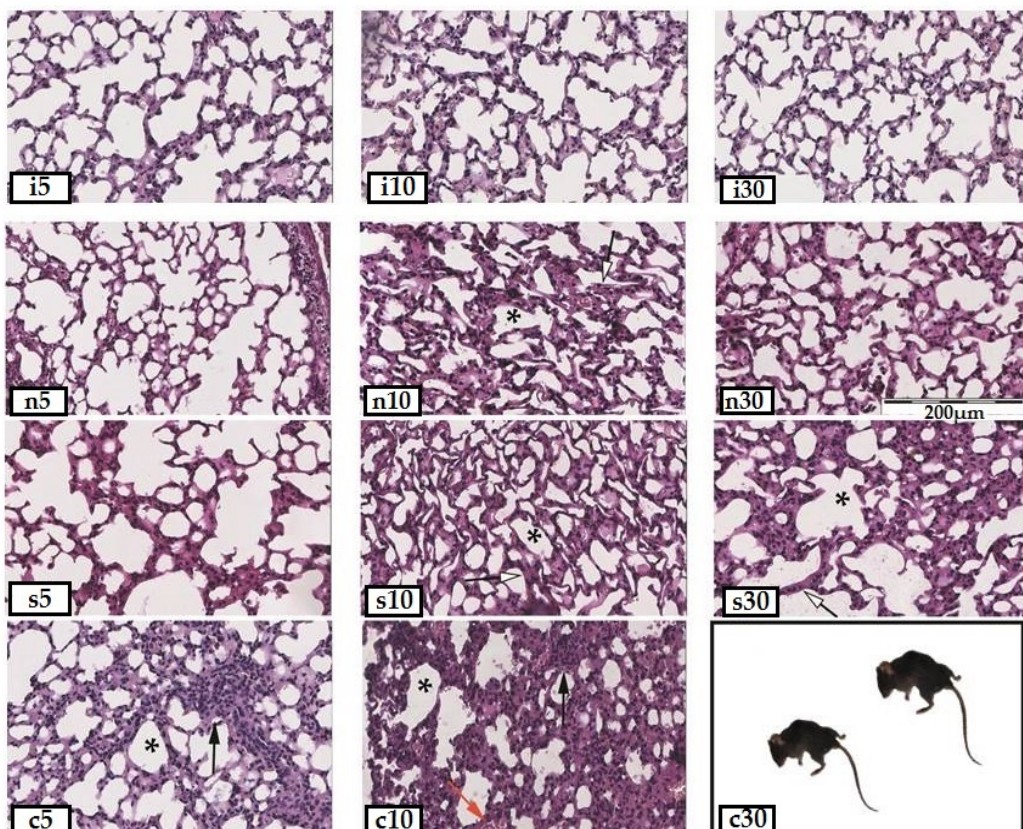

**Figure 2.** Paraffin sections of the lungs of mice from different groups of the experiment (stained with hematoxylin and eosin, 20×: i5—intact healthy animals on the 5th day, i10—on the 10th day, i30—on the 30th day; n5—animals with administration of intranasal vaccine and SARS modeling on the 5th day after infection, n10—on the 10th day, and n30—on the 30th day; s5—animals with administration of subcutaneous vaccine and SARS modeling on the 5th day of infection, s10—on the 10th day, and s30—on the 30th day; c5—animals with SARS modeling on the 5th day of infection, c10—on the 10th day, and c30—on the 30th day; *—areas of atelectasis; black arrow—lymphoid infiltration of the interalveolar septa; red arrow—stasis of erythrocytes in the capillaries; white arrow—thickening of the interalveolar septa.

Of note, in the group with SARS modeling but without treatment, active lymphoid infiltration was observed at the sites of the damaged lungs, while in the groups with the administered vaccine, lymphoid cells (stained by hematoxylin and eosin) were found in the interalveolar septa on average 53.6 ± 3.1% less ($p < 0.05$). In our opinion, as a result of the vaccine administration, B and T cells came into contact with the unique sequences of SARS-CoV-2 genome found in the 'head' regions of the vaccine constructs, and the formation of immunological memory cells occurred. We think that, as a result of the use of the adjuvant oligonucleotide vaccine, the mice's adaptive immunity had already been formed by the time of infection, and the B and T lymphocytes neutralized the viral particles more effectively without pronounced lymphoid infiltration and prevented the development of severe complications of the disease in the form of epithelial damage, interstitial edema, and fibrosis. Logically, the low concentration of lymphoid cells in the interalveolar septa of the vaccinated mice after the SARS modeling can be clearly explained. B and T memory cells regulate and decrease lymphoid infiltration, since the required specific B and T cells are already formed to resist the virus, and there is no need for pronounced inflammation in organisms that are 'ready' for infection, which can be seen in practice in vaccinated organisms. Memory lymphocytes require significantly fewer pre-immune inflammatory mediators and co-stimulatory signals than non-immune lymphocytes to start an immune response to their antigen and can start it without inflammation or with minimal symptoms

of inflammation [28]. Other researchers have also shown a decrease in the number of immune cells in the affected tissue in vaccinated versus unvaccinated animals [29]. On the other hand, it is assumed that the observed antiviral effect, caused by the adjuvant oligonucleotide vaccine, is based on the preactivation of innate immunity cells, which allows them to respond earlier and more intensively to a viral infection. The phenomenon in which the response of innate immune cells to a secondary stimulus is modulated by a previous interaction with the same or an unrelated pathogen has been termed innate immune memory. This is another one of the most important effects of vaccinations [30].

## 4. Discussion

Scientists have conducted experiments over the past few decades and collected sporadic but convincing data on the possibility of using nucleic acids as an active immunogen [31,32]. Still, there are two serious questions that should be addressed.

First, the question of whether dendritic cells can present an antigen fragment of a adjuvant oligonucleotide vaccine containing a unique sequence of the coronavirus RNA genome remains insufficiently studied. Leukocytes and dendritic cells are able to penetrate into all parts of the body and can absorb, transport, process, and present antigens to T lymphocytes [33]. Dendritic cells move to secondary lymphoid organs to provide the delivery of antigens to T lymphocytes, and this, in turn, contributes to the manifestation of a powerful antigen-specific immune response. If B cells are affected when an antigen is detected, then this also applies to T cells, since B cells cannot alone be involved in immune response [34]. This suggests that if B cells are activated, then so are T cells, whose full activation depends on dendritic cells. Currently, there is no single antigen that can activate B cells without activating T cells. However, in systemic lupus erythematosus, B cells are activated, and antibodies are produced that attack the DNA of the cells of a sick person [31,32], but the activation of T cells by dendritic cells due to the antigen presentation of nucleic acid fragments has not yet been shown. In fact, this issue has not yet been closely studied yet.

Second, we assume that the human body can produce antibodies that can enter human cells during viral infection and target particular fragments of nucleic acids in RNA viruses. While generally speaking, to the best of our knowledge, antibodies do not pass easily through intact cellular or subcellular membranes in living cells [35], obviously this is not always the case. Numerous investigations conducted mostly in cultured cells over the years have demonstrated that it is possible to facilitate the cellular internalization of antibodies [36]. The potential for in vivo therapeutic advantages of a nuclear-penetrating lupus anti-DNA autoantibody have also been shown in a number of studies [37,38]. Thus, the effectiveness of adjuvant oligonucleotide vaccines will be determined by the ability of dendritic cells to present antigens of coronavirus oligonucleotide sequences, as well as the ability of antibodies to attack unique coronavirus sequences inside infected host cells.

It should also be taken into account that phosphorothioate oligonucleotides have chiral centers [39], which, of course, are important for their conformational binding to proteins due to sulfide bonds [40]. Research in this area shows that R-P chiral phosphorothioate oligonucleotides elicit a stronger response to their use than S-P oligonucleotides [41–43]. Thus, a more thorough study of oligonucleotide–protein interactions is needed, which will make it possible to obtain adjuvant oligonucleotide vaccines with maximum efficiency [44,45].

## 5. Conclusions

This manuscript offers the results of studies on an adjuvant oligonucleotide vaccine of the La-S-so type for the first time. The data obtained indicate a positive effect of the vaccine on the survival and weight gain of the transgenic mice, as well as on the organometric parameters and tissue structure of the lungs. The observed effects are explained by the activation of both their innate and adaptive immunity. Generally, the best trend was shown by the group for which the intranasal vaccine was administered. In this group, the

weight gain, area of the lumen of the alveoli, and area of vessels were close to those of the intact control.

In the absence of developed test systems capable of assessing the formation of antibodies to the administered oligonucleotide vaccine, the data obtained indicate the prospects for the development of this vaccine platform. The synthesis of adjuvant oligonucleotide vaccines can be automated relatively easily and prepare us for our next encounter with a SARS-CoV-2 coronavirus pandemic. Unfortunately, there is no doubt that it will happen again. Further research will be aimed at proving (1) the existence antibodies capable of penetrating cells and attacking the unique nucleic acid sequences of RNA viruses, and (2) the antigen presentation ability of dendritic cells with respect to nucleic acids. Even though adjuvant oligonucleotide vaccines do not yet fit within the framework of a modern textbook on immunology, they have great potential in prophylaxes of COVID-19 and other diseases caused by coronaviruses and RNA viruses. While the details of the action of adjuvant oligonucleotide vaccines are being investigated in depth, positive outcomes from their use are already apparent.

**Author Contributions:** Conceptualization, V.V.O.; methodology, V.V.O., K.A.Y. and T.P.M.; software, I.A.N.; formal analysis, V.V.O. and K.V.L.; investigation, V.V.O., K.A.Y. and T.P.M.; resources, I.A.N. and A.V.K.; data curation, V.V.O. and K.V.L.; writing—original draft preparation, V.V.O.; writing—review and editing, V.V.O., K.A.Y., O.A.A. and A.I.B.; visualization, V.V.O. and K.A.Y.; supervision, V.V.O.; project administration, V.V.O. All authors have read and agreed to the published version of the manuscript.

**Funding:** The research results are partially obtained within the framework of a state assignment for V.I. Vernadsky Crimean Federal University for 2021 and the planning period of 2022–2023 No. FZEG-2021-0009 ('Development of oligonucleotide constructs for making selective and highly effective preparations for medicine and agriculture', registration number 121102900145-0).

**Institutional Review Board Statement:** Not applicable.

**Informed Consent Statement:** Not applicable.

**Data Availability Statement:** Not applicable.

**Acknowledgments:** We would like to thank our many colleagues, too numerous to name, for their technical advice and lively discussions that have prompted us to write this review. We apologize to our colleagues whose work has not been cited. We are very much indebted to all the anonymous reviewers, our colleagues from the lab on DNA technologies, PCR analysis, and the creation of DNA insecticides (V.I. Vernadsky Crimean Federal University, Department of Molecular Genetics and Biotechnologies), our colleagues from the lab on cell technologies and the creation of DNA medicines (V.I. Vernadsky Crimean Federal University, Department of Molecular Genetics and Biotechnologies), and Olinscide Biotech LLC for their valuable comments on our manuscript. We are very thankful to Georgia Morgan for her English language editing service.

**Conflicts of Interest:** The authors declare no conflict of interest.

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
