# Peer review of "Adjuvant Oligonucleotide Vaccine Increases Survival and Improves Lung Tissue Condition of B6.Cg-Tg (K18-ACE2)2 Transgenic Mice"

_scipharm, doi:10.3390/scipharm91030035_

Round 1

Author Response

Dear Reviewer,

please, see the attachment with our point by point answers to your comments.

Thank you for your help and collaboration.

Kind regards, Dr. V. Oberemok and co-authors.

Reviewer 2 Report

Remarks to the authors: 

The brief report of Oberemoa et al., aims to evaluate the effectiveness of La-S-so oligonucleotide vaccine in an adapted mouse model for SARS-CoV-2 infection. Authors monitored weight loss of vaccinated animals upon challenge with SARS-CoV-2 and analyzed tissue damage in lung. The paper is easy to read and rationally constructed. Nevertheless, some points need to be improved to support the results. 

Comments:

The number of experiments is not noted. The number of replicates in each group does not appear in figure legends.

In the abstract and results part, Authors assume that their oligonucleotide vaccine increase survival rate of animals infected with SARS-CoV-2 but no data support this conclusion. Authors must include the survival curve in the result section.

In figure 1, authors show mass of animals. However, the initial body weight of their group is different. It is therefore difficult to assess the variation in body weight during infection with this representation. Authors should represent their data as percentage of the initial weight.

Legend of figure 1 need to be developed. Positive and negative group are not clearly explained in the legend. We need to search along the manuscript to obtain this information. 

A group vaccinated with a commercial vaccine and a group vaccinated with only CpG or only oligonucleotide backbone are required to truly appreciate the value of developing this kind of vaccine formulation.

Table 1, coma (,) should be replaced by dot (.) for decimal value.

Legend of figure 2 need to be completed. Some arrows (white, red or black) and stars appear on the pictures but no explanation about them on the legend of the figure…

In the discussion, authors suggest that their oligonucleotide La-S-so is immunogenic and play a role in the protection conferred by the vaccine. Effect not only managed by the adjuvant (CpG motif) included into their formulation. Authors must discuss more this subject.

To go further, authors should add a standard antigen like ovalbumin to evaluate the effect of their formulation on antibody response development. 

Author Response

(The authors gave the same response as above.)

Round 2

Reviewer 1 Report

All comments have been adressed, so I recommend that the paper be published in the Scientia Pharmaceutica in its current form.